# Inhaled Placental Mesenchymal Stromal Cell Secretome from Two- and Three-Dimensional Cell Cultures Promotes Survival and Regeneration in Acute Lung Injury Model in Mice

**DOI:** 10.3390/ijms23073417

**Published:** 2022-03-22

**Authors:** Vasily A. Kudinov, Rafael I. Artyushev, Irina M. Zurina, Elena S. Zorina, Roman D. Lapshin, Ludmila B. Snopova, Irina V. Mukhina, Irina N. Saburina

**Affiliations:** 1Laboratory of Cell Biology and Developmental Pathology, FSBSI Institute of General Pathology and Pathophysiology, 125315 Moscow, Russia; rifiraf21ibmx@gmail.com (R.I.A.); izurina@gmail.com (I.M.Z.); saburina@mail.ru (I.N.S.); 2Institute for Regenerative Medicine, Sechenov First Moscow State Medical University (Sechenov University), 8-2 Trubetskaya St., 119991 Moscow, Russia; 3Postgenomic Data Analysis Laboratory, Institute of Biomedical Chemistry, 119121 Moscow, Russia; el.petrenko@bk.ru; 4Central Research Laboratory, Privolzhsky Research Medical University, 603005 Nizhny Novgorod, Russia; r.d.lapshin@gmail.com (R.D.L.); lsnopova@mail.ru (L.B.S.); mukhinaiv@mail.ru (I.V.M.)

**Keywords:** acute respiratory distress syndrome, adult stem cells, cell spheroids, cell-free therapy, COVID-19, extracellular vesicles, inhalation, inflammatory diseases, lung surfactant, tissue regeneration

## Abstract

Acute lung injury (ALI) and acute respiratory distress syndrome (ARDS) is a common clinical problem, leading to significant morbidity and mortality, and no effective pharmacotherapy exists. The problem of ARDS causing mortality became more apparent during the COVID-19 pandemic. Biotherapeutic products containing multipotent mesenchymal stromal cell (MMSC) secretome may provide a new therapeutic paradigm for human healthcare due to their immunomodulating and regenerative abilities. The content and regenerative capacity of the secretome depends on cell origin and type of cultivation (two- or three-dimensional (2D/3D)). In this study, we investigated the proteomic profile of the secretome from 2D- and 3D-cultured placental MMSC and lung fibroblasts (LFBs) and the effect of inhalation of freeze-dried secretome on survival, lung inflammation, lung tissue regeneration, fibrin deposition in a lethal ALI model in mice. We found that three inhaled administrations of freeze-dried secretome from 2D- and 3D-cultured placental MMSC and LFB protected mice from death, restored the histological structure of damaged lungs, and decreased fibrin deposition. At the same time, 3D MMSC secretome exhibited a more pronounced trend in lung recovery than 2D MMSC and LFB-derived secretome in some measures. Taking together, these studies show that inhalation of cell secretome may also be considered as a potential therapy for the management of ARDS in patients suffering from severe pneumonia, including severe acute respiratory syndrome coronavirus 2 (SARS-CoV-2), however, their effectiveness requires further investigation.

## 1. Introduction

Acute lung injury (ALI) is a common condition characterized by upregulation of inflammatory mediators. Dyspnea, severe hypoxemia, and pulmonary edema follow ALI, producing significant mortality rates [1]. Acute respiratory distress syndrome (ARDS), the clinical manifestation of ALI, is a major cause of acute respiratory failure, with a 33–48% mortality rate in critically ill patients [2]. ARDS causes extensive surfactant dysfunction and depletion [3], causing pulmonary structural unit collapse, limited gas exchange, and severely reduced blood oxygen levels, leading to fatal hypoxia [2]. Despite extensive ARDS research, mortality in adult patients remains high [4].

Pandemic respiratory viruses lead to ARDS development more frequently than seasonal viruses [5]. A recent worldwide outbreak of pneumonia caused by severe acute respiratory syndrome coronavirus 2 (SARS-CoV-2) led to a high percent of ARDS, or even death, among adult patients [6].

However, there are still no approved medicines for ARDS, thus, the development of effective treatment strategies or agents is highly desired. Among these, multipotent mesenchymal stromal cell (MMSC) therapies are considered as a promising approach for the treatment of ARDS in different respiratory viral infection models. MMSCs offer new approaches for numerous pathologies [7], including those related to lung diseases [6,8,9,10]. MMSCs possess anti-inflammatory, immunomodulatory, regenerative, proangiogenic, antifibrotic, and antimicrobial properties, and clinical trials are currently underway [2,7,11]. The therapeutic effects of MMSCs are mainly mediated by paracrine activity, which involves the release of bioactive substances collectively known as the secretome [7,11]. The secretome contains extracellular vesicles (EVs), growth factors, cytokines, and chemokines, and is considered a promising cell-free therapeutic agent. Due to clinical data indicating their safety, MMSCs are the preferred source of the secretome and derived EVs [11,12,13,14,15,16,17]. MMSC-derived secretome can be sterilized by filtration and produced as an off-the-shelf product, while MMSCs themselves cannot. Moreover, MMSC-derived secretome are free from the safety issues associated with cell-based therapy [6,7,18,19].

Studies of the secretome and derived exosomes as treatment for lung diseases have mainly used systemic administration via intravenous injection or direct intratracheal administration [11]. However, inhalation, enabled by the small size of proteins and EVs, is a less invasive administration route. Furthermore, due to the large pulmonary surface area (approximately 35–100 m^2^), direct pulmonary delivery provides rapid and prolonged effects [17].

The regenerative capacity of the secretome depends on many factors. For instance, transitioning from monolayer culture to nonadhesive, three-dimensional (3D) culture can dramatically change the secretome contents and therapeutic potential [20,21]. Therefore, we aimed to develop lyophilized biomimetic formulations containing secretome from two-dimensional (2D) and three-dimensional (3D) cultures of placental MMSCs and to assess their therapeutic potential against ALI in vivo. We also used the secretome from 2D- and 3D-cultured lung fibroblasts (LFBs) as an active comparator, which effectively attenuates bleomycin-induced lung fibrosis [22]. The flow chart of the study is shown in Figure 1.

## 2. Results

### 2.1. 2D and 3D Cell Cultures

Placenta-derived MMSCs that were previously isolated and characterized were used in this study [23].

The MMSCs were isolated from term human placenta obtained after Caesarian section. The enzymatic approach allowed for rapid cell isolation—they readily adhered to culture plastic on the next day after tissue digestion and reached 100% confluent monolayer within several days. At passage 3, they exhibited high proliferative activity, as numerous doubling cells were observed in culture, and were characterized by a spindle-shaped morphology (Figure 2A). Flow cytometry of passage 3 cells revealed negative endothelial and blood cell marker expression (CD14, CD31, CD45, CD34, CD11b, and CD19 < 1%) (Figure 2E–G) and high expression of surface markers characteristic of MMSCs population (CD44, CD29, CD90, CD105, and CD73 > 85%) (Figure 2H–L). After confirming the MMSC phenotype of the culture, cells were used for the obtainment of conditioned media.

### 2.2. Protein Composition of Cell Secretome

To understand the molecular mechanism mediating the regenerative and protective properties of our secretome-based formulations, we estimated proteomic composition by liquid chromatography–tandem mass spectrometry (LS/MS/MS).

In secretomes from 2D and 3D MMSC cultures, 281 and 286 proteins were identified, respectively. Only 234 proteins were shared by the 2D- and 3D-cultured MMSC secretomes (Figure 3A). Of these shared proteins, 23.4% and 43% were annotated as cytoplasmic proteins for 2D- and 3D-cultured MMSC secretomes, respectively (Figure 4A,B). Furthermore, 23.1% and 9.5% of the shared proteins were annotated as “extracellular region” or “secreted” for 2D- and 3D-cultured MMSC secretomes, respectively (Figure 4A,B). Annotation of the 47 proteins exclusive to the 2D-cultured MMSC secretome indicated their participation in 22 pathways, including growth factor signaling pathways, the endothelium signaling pathway, the hypoxia response, immune cell activation, angiogenesis, and cytoskeletal regulation (Figure 3C). Annotation of the 52 proteins exclusive to the 3D-cultured MMSC secretome indicated their involvement in only 13 pathways, including FGF, TGF-β, PDGF, the IFN-γ signaling pathway, angiogenesis, and cytoskeletal regulation (Figure 3D). Molecular functions for proteins exclusive for 3D-cultured MMSC secretome were more diverse compared with 2D MMSC culture (Figure 4E,F). The biological process analysis results for MMSC secretome proteins were similar (Figure 5A,B).

2D- and 3D-cultured LFB secretomes contained 319 and 306 proteins, respectively, and shared 236 proteins (Figure 3B). Of the shared proteins, 40% and 24.3% were annotated as cytoplasmic proteins for 2D- and 3D-cultured LFB secretomes, respectively, and 22% and 11.4% were annotated as “extracellular region” or “secreted” for 2D and 3D, respectively (Figure 4C,D). These observations align with data from a previous study of the lung spheroid cell secretome [22]. Annotation of the 83 proteins exclusive to the 2D-cultured LFB secretome indicated their participation in 33 pathways, including growth factor signaling pathways, the morphogenic Wnt signaling pathway, the endothelium signaling pathway, the hypoxia response, immune cell activation, inflammation, blood coagulation, angiogenesis, and cytoskeletal regulation (Figure 3E). The 70 proteins exclusive to the 3D-cultured LFB secretome were involved in only nine pathways, including Wnt, Notch, and TGF-β signaling (Figure 3F). Molecular functions and biological processes for proteins exclusive to 3D-cultured LFB secretome were close to secretome proteins from 2D LFB culture (Figure 4G,H and Figure 5C,D).

### 2.3. ALI Model

The survival rates of all six groups were monitored daily. The highest mortality rate (40%) was observed in the nontreated group with ALI (ALI + NaCl). All treatment groups and the control intact group exhibited 0% mortality throughout the experiment (Figure 6A).

On day 8, the lung weight in the ALI control group (ALI + NaCl) and in groups treated with formulation from LFB-2D culture was significantly larger than in the control group. This change was likely a characteristic of the stage of ALI development. The lung weights were restored almost to their initial values in the groups that received formulations from LFB-3D culture and MMSC cultures (MMSC-2D, MMSC-3D) (Figure 6B).

### 2.4. Histological Examination

NaCl solution (0.9%) did not affect the lung structure, which featured rounded, well-spread alveoli with thin walls and clear lumens (Figure 6C, blue arrows). The alveolar epithelium was characterized by distinct nuclei and oxyphilic cytoplasm. Bronchioles and bronchi were lined with cubic and cylindrical epithelium, and lumens were clear (Figure 6C, green arrow).

ALI development on day 8 was accompanied by moderate edema of the loose connective tissue surrounding the bronchi and accompanying vessels. Diffuse infiltration of connective tissue around bronchi with inflammatory elements (Figure 6D, blue arrows) and areas of desquamation of the bronchial mucosa prismatic epithelium were observed (Figure 6D, green arrow). Mucus production by goblet cells was enhanced (Figure 6D, green arrow), and the lung experienced incomplete collapse, producing alveoli lumens with irregular shapes and sizes (Figure 6E, blue arrow). Interalveolar septa were thickened due to edema of the alveolar septa stroma (Figure 6E, green arrow). Part of the alveoli contained exudate with an admixture of erythrocytes, fibrin, and desquamated alveolar epithelium (Figure 6D, orange arrow). We also observed a plethora of blood clots in some vessels of the vascular bed (Figure 6E, orange arrow). Exudate score and interstitial edema and hyaline membrane formation score are shown in Figure 6J,K.

2D-LFB-Sec inhalation resulted in no peribronchial edema; bronchi and small bronchioles had open, clear lumens. However, most alveoli exhibited incomplete expansion (Figure 6F, blue arrows) and partial wall thickening (Figure 6F, green arrows). The exudative score was significantly lower compared to the control ALI group (Figure 6J). The interstitial edema and hyaline membrane formation score did not differ from the ALI control group (Figure 6K). 3D-LFB-Sec also significantly reduced ALI severity, as observed on day 8. There was no interstitial edema around the bronchi and associated vessels; bronchi, small bronchioles, and some of the alveoli featured an open, clear lumen (Figure 6G, blue arrows). However, some areas also exhibited reduced lumen size and uneven alveoli contours (Figure 6G, green arrows). The exudative score and the interstitial edema and hyaline membrane formation score were significantly lower compared to the control ALI group (Figure 6J,K).

2D-MSC-Sec treatment relieved ALI on day 8 after LPS administration. An absence of peribronchial edema was observed, and the bronchi and small bronchioles had open, clear lumens (Figure 6H, blue arrow). The majority of the alveoli were inflated with clear lumens and thin walls (Figure 6H, green arrows); only isolated, small areas had unevenly thickened alveolar walls (Figure 6H, orange arrow). The exudative score and the interstitial edema and hyaline membrane formation score were significantly lower compared to the control ALI group and were lower compared to groups treated with secretome from LFB cultures (and Figure 6J,K).

3D-MSC-Sec inhalation resulted in no interstitial edema around bronchi and associated vessels, and bronchi and small bronchioles featured clear, open lumens (Figure 6I, blue arrows). The alveoli were mainly straightened with clear lumens and thin walls (Figure 6I, green arrows). Only small areas had unevenly thickened alveolar walls and incompletely expanded alveoli (Figure 6I, orange arrow). The assessment of exudation and edema has shown that 3D-MSC-Sec treatment is associated with the lowest values of scoring compared to the control ALI group (adjusted *p* < 0.0001) (and Figure 6J,K). Additional representative histological areas are presented in the Appendix A.

### 2.5. Pulmonary Coagulation and Fibrinogen/Fibrin Deposition 

Immunohistochemical staining of lung tissue sections revealed fibrinogen deposits in all studied groups, including healthy control animals (Figure 7A). The percentage of fibrinogen-positive regions in the control group lung tissue was 6.66%. The highest percentage of fibrinogen expression (28.62%), with pronounced edema of the connective tissue and thickening of the interalveolar septa, was observed in the nontreated ALI control group. Repeated inhalation of all investigated formulations resulted in 7–11% fibrin/fibrinogen expression by day 8. This expression was significantly lower than that observed in the control ALI group, but significantly higher than that in the control group (Figure 7B). No differences were observed between the MMSC-2D and MMSC-3D groups.

## 3. Discussion

The current consensus is that the MSC secretome is their main mode of action. Preclinical studies of lung diseases have demonstrated that the MMSC secretome can effectively treat chronic obstructive pulmonary disease, ARDS, idiopathic pulmonary fibrosis, and severe pneumonia [14,22,24]. In most studies, MMSCs derived from bone marrow, Wharton’s jelly of the umbilical cord, and adipose tissue have been used as the source of the secretome or EVs. Placenta is also an available postpartum source of MMSCs that involves no ethical controversies regarding its use for secretome production in clinical practice [25]. Profiles of proteomes secreted by MMSCs from different sources and cultured under the same conditions share similar functional features, however, proteomes of postpartum MMSCs from the umbilical cord and placenta have been predicted to have greater therapeutic potential [26].

Although soluble factors secreted by viable MMSCs into the culture medium offer an attractive cell-free therapeutic alternative, to our knowledge, their regenerative potential has not yet been fully investigated. Therapeutic properties of secretome/exosomes were reported to be modulated by factors related to their culture condition such as cell source, oxygen tension (normoxia vs. hypoxia), growth factor composition, and physical microenvironment [27]. Most studies produced conditioned media (CM) in monolayer culture, but several studies used different 3D cultures, in particular, cell spheroids. Spheroid cultures need a spatial handling but yield more cells compared to conventional monolayer cultures, and thus more secreted factors [28]. Three-dimensional cell culture enables better recapitulation of the in vivo environment and was proven to be beneficial for MMSC culture in terms of enhancing their therapeutic potential [27]. In addition, cells located at the center of the spheroid may be in a relative hypoxic condition compared to cells on the surface, which leads to increase of certain pro-regenerative paracrine signaling molecules and immunomodulatory factors [28,29]. The therapeutic effects of MMSC from 3D culture were also translated to their secretome [28,30]. 

We have developed an original cell spheroid culturing protocol in our lab. To provide an appropriate nursing ecosystem for MMSC, the fetal calf serum (FCS) and basic fibroblast growth factor (bFGF) were added to the basic growth medium. We have not used a serum-free culture medium and serum starvation because they may cause changes in characteristics of cells and the production yield and properties of exosomes. Specifically, serum deprivation may induce cell death (in MMSC) [31,32,33]. It was shown that when serum concentrations are reduced from 10% to 1%, the size distribution, total quantity, and protein composition of in vitro–derived EVs were different [34]. Depletion of fetal serum from EVs may also negatively effects on secretome content. It is reported that in response to the various depletion process, EV-depleted fetal serum may influence parent cell phenotype and possibly their qualitative and quantitative production of EVs [31,35]. To minimize the influence of fetal serum EVs and growth factors, we conditioned cell cultures over 72 h. Our internal lab data showed that cells actively consume bFGF from the full growth medium, decreasing their concentration from 20 ng/mL to 334 pg/mL after 72 h for 3D MMSC cultures (unpublished data) and 331 pg/mL for 2D cultures [23]. These data align with other studies showing that cells actively uptake soluble serum components and EVs for their growth and survival [36]. Therefore, we chose not to use unconditioned growth medium as a control, but compare the therapeutic potential of CM from 3D-cultured MMSC obtained during spheroid formation and compactization stage and in parallel from monolayer MMSC culture to CM from 2D- and 3D-cultured LFB as a control. The culture conditions were equivalent for all cells in our study.

The administration route may influence the biodistribution of secretome products and their final therapeutic effect. Monsel et al. showed that microvesicles (MVs) released by human bone marrow–derived MSCs reduces the severity of endotoxin-induced ALI in mice by transferring keratinocyte growth factor (KGF) mRNA to the injured alveolus. However, the MVs were administered intravenously in this study [24]. Another study showed that inhalation of lung spheroid cell secretome and exosomes promotes lung repair in pulmonary fibrosis more effectively than inhalation of 2D-cultured MSC secretome and exosomes. Interestingly, recent studies have indicated that exosomes reproduce only part of the regenerative potency of the full secretome from which they are isolated [22]. 

Here, we demonstrated that inhalation of lyophilized secretome-based formulations improved survival in a lethal endotoxin-induced ALI mouse model. We compared formulations containing secretome derived from either 2D and 3D placental MMSC cultures or 2D and 3D LFB cultures, in terms of their regenerative capacity. ALI significantly damaged the lung structure. Inhalation of CM from 2D and 3D LFB cultures resulted in partial lung structure restoration; peribronchial edema was not observed in either group, and the bronchi and small bronchioles had clear, open lumens. However, there were signs of incomplete alveoli expansion and slight wall thickening in both LFB groups. The interstitial edema and hyaline membrane formation score were significantly higher in LFB-2D group. 

Administration of 2D and 3D MMSC formulations relieved ALI on day 8 after LPS inhalation. Experimental groups treated with MMSC secretome–based formulations exhibited almost full restoration of the lung architecture to a healthy level after three inhalations. Only small areas with unevenly thickened alveolar walls and incompletely straightened alveoli were noted. These findings align with those of previous studies in rats that investigated the effects of systemically administered MSCs and MSC exosomes [24,37]. These studies have shown that secretome-based products can confer a level of protection and regeneration similar or superior to that conferred by the cells themselves. Secretome from stem cells contains signaling molecules and recyclable materials for survival, proliferation, and differentiation of recipient cells. It was shown that MMSC secretome provides several stimulatory and inhibitory bioactive factors at variable concentrations that might sustain physiological kinetics in the local microenvironment [38]. Potential mechanisms underlying the beneficial effects of MMSCs include the following: an increase in human KGF protein levels in injured alveoli that promotes alveolar epithelial cell repair; an immunomodulatory effect on monocytes and alveolar microphages that suppresses cytokine-induced lung injury and lung protein permeability; enhanced alveolar epithelial type 2 cell metabolism via delivery of key metabolic enzymes (glyceraldehyde 3-phosphatase dehydrogenase, pyruvate kinase); a transfer of mitochondria, increasing the bacterial clearance; restoration of vascular endothelial permeability via release of VEGF and HGF; and protection of endothelial cells against apoptosis [2,24,39]. However, aggregation and 3D spatial organization of MMSCs into multicellular spheroids reportedly promotes their anti-inflammatory properties through increased anti-inflammatory factor secretion [20,21,40,41,42]. Our data have also indicated that 3D-MSC-Sec inhalation has a more pronounced restorative trend in damaged lung structure compared to other formulations, according to the exudative score and the interstitial edema and hyaline membrane formation score. At the same time, no significant differences were found between groups treated with the secretomes from 2D MMSC, 3D MMSC, and 3D LFB cultures. These findings can be partially explained by differences in the composition of the secretome obtained from 3D cell cultures, which potentially may be more effective in restoring disturbed tissue homeostasis. Differences in the contents of exosome RNA cargo in the secretome may also explain differences in treatment capacity [22]. The complexity of cell secretome content indicates that more studies are needed to fully understand the mechanisms of their biotherapeutic activity.

Our study also assessed the effect of secretome inhalation on fibrin deposition in the lungs. Nearly all patients with ALI demonstrate abnormalities in alveolar fibrin turnover. These abnormalities vary from subtle changes in molecular coagulation and fibrinolysis markers to more evident fibrin deposition in smaller airways. Fibrin has a key role in host defense, and fibrin depositions activate neutrophils and fibroblasts and decrease alveolar fluid clearance, thereby inactivating surfactant and favoring alveolar collapse, increasing pulmonary dead space, and causing additional endothelial injury [43]. In our study, all secretome-treated groups showed significantly less fibrin deposition than the ALI control group on day 8. However, these results need further clarification, as they may be related to residual heparin content in CM after 72 h of conditioning.

This work had several limitations. First, the results of the mouse ALI models may not be easily translated into studies of complex and heterogeneous human populations. Our models also relied on a single, high-dose administration of LPS to healthy animals; however, ALI is usually associated with pre-existing risk factors and develops over a longer period. The lung injury improvement was evaluated solely by histological examinations in this study. It would be better to demonstrate lung injury improvement by measuring cytokines in lung tissue and bronchoalveolar lavage fluid or by immunohistochemical assessment of other markers (recovery of alveolar epithelial cells I and II, the activity of antioxidant enzymes, pulmonary surfactant recovery). The frequency of agent administration may have been another limitation. We used only one dose of cell secretome, an arbitrary choice guided by previous experience. More frequent or continuous administration may be more effective. Future studies will include a dose–response assessment to determine the optimal dosage regimen. In our study, treatment was delivered via inhalation using a nebulizer. An investigation into different administration routes to determine the optimal delivery method and frequency of administration would help maximize benefits. The last important limitation is the presence of residual proteins and EVs from FCS in the studied CM, which could potentially distort and even reduce the therapeutic efficacy of the MMSC secretome [44]. However, we used identical cell culture conditions, so we assume that the contribution of FCS-derived EVs was the same for all cell cultures. Prolonged conditioning leads to most of the EVs in the secretome originating from MMSCs. This enrichment of the CM with EVs derived from MMSCs should potentiate its therapeutic potential. This is consistent with recent studies showing that the regenerative potential of EVs from fetal serum is significantly lower than vesicles from MMSCs [45]. Interestingly, EVs collected from the MMSC cultured in fetal serum–containing medium had greater wound healing and angiogenic effects for endothelial cells compared with FBS-derived EVs [44]. Based on the literature, the MMSC secretome examination may benefit from the use of chemically defined, serum-free, and xeno-free medium that is not only optimized for cell growth and viability for a variety of cell types, but is also free of exogenous contaminating fetal serum-derived EVs and extracellular protein/RNA species [31]. Lastly, we presented here the comparison of the proteomic composition of MMSC and LFB secretomes from 2D and 3D cultures to confirm their differences. It is obvious that the secretome composition may influence biotherapeutic properties and potential for application. In addition to proteins mRNA, miRNA, and lipid content may play a significant role. We acknowledge that it would be beneficial to determine which proteins and RNA are the important functional components determining the therapeutic effects of the secretome. Future experimental directions may include MSC preconditioning [46], cell bioengineering to modulate the secretome profile [47,48], and developing subsequent purification and standardization methods for scaling up secretome production [11]. 

In summary, we demonstrated that inhaled formulations based on lyophilized CM derived from 2D and 3D placental MMSC and LFB cultures improved survival in mice with LPS-induced ALI. Cell-free secretome-based formulations from 2D and 3D cell cultures were effective and safe in the lethal ALI model, suggesting a possible noninvasive alternative to stem cell-based regenerative therapy. Lyophilization preserves proteins and vesicles and prolongs storage without special requirements, which means that secretome-based products can be stored as an off-the-shelf powder [49]. Further experimental and clinical studies are needed to elucidate the fundamental mechanisms that mediate the protective effects of MMSC secretome from 2D and 3D cell cultures in the settings of ALI.

## 4. Materials and Methods

### 4.1. Chemicals and Reagents

The following reagents were purchased for the study. Anti-CD105/anti-CD90/anti-CD73 (562245, BD Bioscience, San Jose, CA, USA), anti-CD44 (130-113-897, Miltenyi Biotec, Auburn, CA, USA), anti-CD29 (130-101-275, Miltenyi Biotec, USA), anti-CD14 (130-113-709, Miltenyi Biotec, USA), anti-CD31 (130-110-806, Miltenyi Biotec, USA), and the mix of anti-CD45/anti-CD34/anti-CD11b/anti-CD19 (130-125-285, Miltenyi Biotec, USA) were purchased for cell immunophenotyping. Universal antigen retrieval reagent (ab208572, Abcam, Cambridge, UK), hydrogen peroxide blocking solution (ab64218, Abcam), protein blocking solution (ab64226, Abcam), primary rabbit polyclonal antibodies against fibrinogen (ab34269, Abcam), rabbit-specific IHC polymer detection kit HRP/DAB (ab209101, Abcam), DAB chromogen (ab64238, Abcam), hematoxylin (ab220365, Abcam), hematoxylin and eosin (ab245880, Abcam), TBS solution (ab64204, Abcam) for immunohistochemistry. Hematoxylin and eosin (ab245880, Abcam) were purchased for histology. Trypsin solution (BioLot, Saint-Petersburg, Russia), Versene solution (BioLot), Hank’s Balanced Salt Solution (BioLot), Petri dishes (Corning-Costar, Tewksbury, MA, USA), Dulbecco’s Modified Eagle Medium/Nutrient Mixture F-12 (DMEM-F12; BioLot), glutamine (BioLot), gentamicin (PanEco, Moscow, Russia), insulin-transferrin-selenium (ITS-G; BioLot), basic fibroblast growth factor (bFGF; ProSpec, Ness-Ziona, Israel), heparin (PanEco), and foetal calf serum (FCS; HyClone, South Logan, UT, USA) were purchased for cell culturing. Phosphoric acid (Sigma, Hamburg, Germany), methanol (J.T.Baker, Netherlands), urea (Sigma, Germany), sodium chloride (Fluka-Honeywell, Seelze, Germany), sodium deoxycholate (Sigma, Milan, Italy), acetonitrile (Carlo Erba, Val de Reuil, France), triethylammonium bicarbonate (Sigma, Switzerland), tris(2-carboxyethyl)phosphine (TCEP; Sigma, St. Louis, MO, USA), 4-vinylpyridine (Aldrich, Gillingha, UK), isopropanol (Fisher Chemical, Loughborough, UK), trypsin (Promega, Madison, WI, USA), acetic acid (Carlo Erba, France), ethyl acetate (Carlo Erba), acetonitrile (Carlo Erba), formic acid (Sigma, Germany) were purchased for proteomic analysis of the cell secretome. Maltose monohydrate suitable for use as excipient (Merk KGaA, Darmstadt, Germany) was purchased for secretome lyophilization.

### 4.2. Cell Sources

The study was conducted using the primary cultures of human placenta-derived MMSCs and LFBs (Cell Application, Inc., San Diego, CA, USA) Placenta tissues were collected from three donors (age 23–36) undergoing Caesarean sections at 39–41 weeks of gestation, after receiving the patients’ written, informed consent. All these procedures were performed under aseptic conditions and were approved by The Local Ethical Committee of the FSBSI “Institute of General Pathology and Pathophysiology” and performed in accordance with the Helsinki Declaration. 

### 4.3. Animals

Sixty 8-week-old male C57BL/6 mice were purchased from the Federal State Budgetary Institution of Science’s Scientific Center for Biomedical Technologies of the Federal Medical and Biological Agency (Andreevka, Russia). Animals were individually housed in stainless steel cages in an air-conditioned room (22 ± 1 °C, 55 ± 5% humidity) with a 12 h/12 h light/dark cycle and ad libitum access to food and water. All experiments were conducted with an effort to minimize suffering and the number of animals used.

### 4.4. 2D MMSC and LFB Cell Culturing and Secretome Collection

Placenta-derived MMSCs that were previously isolated and characterized were used in this study [23]. Placental tissue transported to the laboratory was first washed in Hank’s solution containing 100 U/mL gentamycin for 12 h and was then mechanically cut into fragments of no more than 3 mm. Individual cells were isolated from the placental tissue pieces by incubation for 30 min in 0.15% collagenase type II solution (Sigma, Germany) at 37 °C with constant stirring.

Hank’s solution was added to the isolated cells to reduce the collagenase activity. The resulting suspension was filtered through a 100 μm pore filter (Becton Dickinson, San Jose, CA, USA) and centrifuged at 300× *g* for 10 min. The cell pellet was resuspended in full growth medium and plated in Petri dishes at the density of 3 × 10^5^ cells/cm^2^ under standard conditions (37 °C in 5% CO_2_). The full growth medium consisted of basal DMEM-F12 with 2 mM glutamine supplemented with 40 U/mL gentamicin, 1% 100× ITS-G, 20 ng/mL bFGF, 15 U/mL heparin, and 10% FCS. The medium was changed 2–3 times per week, and visual inspection of the culture was performed under a Primovert phase-contrast microscope (Zeiss, Jena, Germany). The cell cultures were passaged at a 1:3 ratio upon reaching 80–85% confluence using Versene and 0.25% trypsin solution. For the LFB cells, the culture conditions were equivalent.

At passage three, the placental MMSCs and LFBs were cultured for 72 h without changing the medium. After 72 h, 500 µL of the resulting CM was collected individually from every dish for further content analysis. The remainder of the CM was pooled together for each cell culture in 50 mL tubes (10 mL of CM from 2 × 10^6^ cells), filtered through 0.22 µm filter to remove any cells and cell debris. The filtered secretome was frozen and stored at −80 °C. Cells at passage 3 were used for characterization and 3D culturing. 

### 4.5. 3D MMSC and LFB Cell Culturing and Secretome Collection

Spheroids were obtained and cultured in nonadhesive multi-well plates prepared from a 2% agarose solution (A-6013, Sigma-Aldrich, Germany) on DMEM/F12 basal medium supplemented with 100 U/mL gentamicin using silicone 3D PetriDish^®^ molds (Microtissues™, USA). Each agarose plate was placed in the well of a 12-well culture plate (Corning-Costar, Cambridge, MA, USA).

Placental MMSCs and LFBs at the third passage were treated with Versene and 0.25% trypsin solutions (BioLoT). The cell suspension was transferred to 15 mL tubes and centrifuged (7 min, 400× *g*). The resulting pellet was resuspended in the complete growth medium to a concentration of 3.3 × 10^6^ cells/mL. Next, 150 μL of that suspension was transferred to nonadherent agarose plates. After an hour, 2 mL of complete growth or induction medium was added to the wells. 

Spheroids from both placental MMSCs and LFB were cultured for 72 h without changing the media. After 72 h, 500 µL of the resulting CM were collected separately from every agarose plate for further content analysis. The remainder of the CM was pooled together for each cell culture in 50 mL tubes, filtered through 0.22 µm filter to remove any cells and cell debris. The filtered secretome was frozen and stored at −80 °C. Every microplate containing 256 spheroids or 0.5 × 10^6^ cells allowed us to collect 2.5 mL of CM, which was equivalent to the proportions described for monolayer cultures.

### 4.6. Immunophenotyping of Human Placental MMSCs

Placental MMSCs were subjected to immunophenotyping after 3 days of passage using the following surface marker profile that is characteristic of MMSCs: CD105, CD90, CD73, CD44, CD29, CD31, CD14 and CD45, CD34, CD11b, CD19 together in a premixed form [50,51]. Specifically, the cell suspension obtained by treatment of the monolayer with Versene and 0.25% trypsin was transferred to 15 mL tubes and centrifuged (7 min, 400× *g*). The pellet was resuspended in phosphate-buffered saline (pH 7.4) containing 1% serum and incubated in the dark (15 min, 25 °C) with antibodies (10 μL of antibodies per 1 × 10^6^ cells) conjugated with fluorescent labels (fluorescein isothiocyanate (FITC), phycoerythrin (PE), Peridinin chlorophyll protein-Cyanine5.5 (PerCP-Cyanine5.5) and allophycocyanin (APC)). The stained cells were centrifuged (5 min, 400× *g*), and the pellet was resuspended in 1 mL phosphate-buffered saline containing 1% FCS in tubes for flow cytometry. The samples were analysed on a Sony SH800 flow cytometer (Sony, Japan).

### 4.7. Proteomic Analysis of the Cell Secretome

#### 4.7.1. Sample Preparation

One hundred microlitres of each sample was transferred into clean tubes and 6 μL of 85% phosphoric acid (up to 5% final concentration; Sigma, Germany) was added and mixed. Then, 800 μL of methanol (J.T.Baker) was added and the samples were mixed again. The resulting suspension was centrifuged at 10,000× *g* at 15 °C for 10 min (Centrifuge 5424R, Eppendorf, Germany). The precipitate was reconstituted in 50 μL of a denaturing solution consisting of 5 M urea (Sigma, Germany), 1% sodium deoxycholate (Sigma, Italy), 300 mM sodium chloride (Fluka-Honeywell), 10% acetonitrile (Carlo Erbo), 100 mM triethylammonium bicarbonate (pH 8.2–8.5) (Sigma, Switzerland), and up to 10 mM freshly added neutralized TCEP (Sigma, St. Louis, MO, USA). The reconstituted denatured protein was incubated at 45 °C for 30 min with constant vigorous stirring at 1200 rpm (Thermo Mixer, Eppendorf, Germany). Then, 6 μL of 2% stabilized 4-vinylpyridine (Aldrich, UK) in 30% isopropanol (Fisher Chemical) was added to a final concentration of 0.2%. The alkylation reaction was incubated in the dark at 20 °C for 20 min. The sample volume was then increased to 500 μL by adding 384 μL of 75 mM triethylammonium bicarbonate (pH 8.2) and thoroughly mixed.

To each sample, 400 ng of trypsin was added from a 100 ng/μL stock (Promega) in 30 mM acetic acid (Carlo Erba) and incubated for 3 h at 40 °C with intermittent stirring (stirring at 1700 rpm for 90 s every 10 min). Then, an additional aliquot of 400 ng trypsin (100 ng/μL) in 30 mM acetic acid was added and the reaction was incubated for 2 h at 42 °C with intermittent stirring as above. At the end of the incubation, 10 μL of absolute formic acid was added to precipitate the reduced deoxycholic acid. The resulting suspension was centrifuged at 12,000× *g* and 5 °C for 10 min. To remove the residual deoxycholic acid, an equal volume of ethyl acetate (Carlo Erba) was added to 500 μL of the supernatant and vigorously stirred for 3 min at room temperature. Then, the mixture was centrifuged at 10,000× *g* for 5 min at 20 °C and then incubated at −20 °C for 10–15 min. The samples were removed from the freezer, the surface organic layer was decanted, and 150 μL of acetonitrile (Carlo Erba) was added to the lower aqueous layer containing the peptides. The mixture was centrifuged at 13,000× *g* for 10 min at 20 °C, and the supernatant was collected and dried under vacuum at 30 °C for 60–70 min with chamber ventilation every 15 min (Concentrator Plus, Eppendorf, Germany). The resulting dry residue was reconstituted in 20 μL of 0.5% formic acid (Sigma, Germany).

#### 4.7.2. High-Performance Liquid Chromatography-Mass Spectrometry (HPLC-MS)

The analysis was performed using a Xevo™ G2-XS QToF quadrupole time-of-flight mass spectrometer (Waters, UK) coupled with an Acquity™ HPLC H Class Plus chromatography system (Waters). The analysis was carried out in positive electrospray ionization mode with increased sensitivity and a normal dynamic range of measurement. The emitter voltage was 3 kV, the drying gas rate was 680 L/min, the focusing gas rate was 50 L/min, the temperature of the ionization source was 150 °C, and the temperature of the desolator was 350 °C. The voltage across the focusing cone was 67 V with a bias of up to 130 V. The ions were recorded in hybrid data-independent acquisition (DIA) MSe-SONAR mode. Specifically, an initial DIA MS scan was performed from 100–1500 *m/z*, followed by a SONAR scan with mass isolation with a quadrupole from 400–1100 *m/z* and an isolation peak width of 22 Da. The time for one complete scan cycle was set at 0.418 s. Fragmentation was carried out in two-phase mode: phase 1—low-energy collision-induced dissociation (CID) fragmentation with argon at 6 eV; phase 2—high-energy ranked CID fragmentation with argon from 15–40 eV. During the analysis, active mass correction (*m/z* = 556.27) with a low activation energy (9 eV) was performed using a leucine enkephalin standard (50 pg/mL in 50% acetonitrile with 0.1% formic acid). The standard was injected into the ionization source for 30 ms at 5 μL/min every 45 s and isolation within 200 mDa.

Chromatographic separation was performed on an Acquity™ UPLC BEHC18 column (1.7 μm particle size, 2.1 × 50 mm column size; Waters) at a flow rate of 0.2–0.3 mL/min and constant temperature of 40 °C. Gradient elution was used, consisting of mobile phase A (aqueous solution of 0.1% formic acid and 0.03% acetic acid) and mobile phase B (solution of 0.1% formic acid and 0.03% acetic acid in acetonitrile) with the following elution scheme: 0–1.5 min, 3% B; 1.5–26.5 min, increased to 19% B; 26.5–42 min, increased to 32% B; 42–43.5 min, increased to 97% B; held in isocratic mode until 47.5 min; decreased to 3% B until 49 min; and held in isocratic mode until 53 min. The flow rate from 43.5–47.5 min was 0.3 mL/min; at all other times, the flow rate was 0.2 mL/min.

#### 4.7.3. Data Analysis

Raw data files were handled using the PLGS (Protein Lynx Global Server, version 3.0.3, Waters, the UK) search engine. Protein search was performed against human amino acid sequences database obtained from the UniProt KB (release May 2021) as a FASTA file with automatically generated reversed concatenated decoy sequences to estimate a false positive rate. Precursor ion mass tolerance was set at 20 ppm (±10 ppm tolerance window) and fragment ion mass tolerance was set to 8 mDa (±4 mDa tolerance window). S-pyridilethylation was included as a fixed modification, whereas oxidized methionine (oxM), deamidated glutamine(dQ), and asparagine (dN) were variable modifications. The minimal peptide length was fixed to eight amino acids, with only one allowed internal missed cleavage. The false discovery rate (FDR) at 1% was determined for peptide and protein identification by accumulating the reverse database hits.

The listed secreted proteins differing in the samples were classified using PANTHER (Protein Analysis THrough Evolutionary Relationships, htpp://pantherdb.org) (21 June 2021) to study molecular functions, cellular components, and pathways. For subcellular classification, the protein subcellular localization predictor mGOASVM (human) (21 June 2021) was used [52].

### 4.8. Preparation of Lyophilized Secretome Based Formulations

First, 50 mL of collected secretome was mixed with cryoprotective agent (maltose), passed through filter with a pore size of 0.22 µm, aliquoted (10 mL of secretome per bottle), and frozen at −80 °C for 3 h. Then samples were lyophilized using a Martin Christ Alpha 1-2 LDplus machine (Martin Christ, Osterode am Harz, Germany) for 40 h. At the end of sublimation, the samples were sealed under a vacuum and rolled up. 

### 4.9. ALI Model

A lethal ALI model in mice was induced in 50 male C57BL/6 mice by inhalation of 10 mg/kg LPS. Another 10 mice were used as intact control. Experimental groups were the following:Control intact (*n* = 10); 0.9% NaCl inhalation;Control ALI (*n* = 10); ALI mice, 0.9% NaCl inhalation;2D-LFB-Sec (*n* = 10); ALI mice treated with LFB-2D formulation;3D-LFB-Sec (*n* = 10); ALI mice treated with LFB-3D formulation;2D-MSC-Sec (*n* = 10); ALI mice treated with MSC-2D formulation;3D-MSC-Sec (*n* = 10); ALI mice treated with MSC-3D formulation.

The experimental animals were subjected to inhalation of the secretome formulations (2D-LFB-Sec, 3D-LFB-Sec, 2D-MSC-Sec, and 3D-MSC-Sec) dissolved in 0.9% NaCl solution (1 mL of reconstituted formulation per mouse) at 30 min, 24 h, and 48 h after LPS administration. Ten control animals not exposed to LPS were subjected to inhalation of 0.9% NaCl.

For both LPS and drug inhalation, a dynamic method of exposure was used when the concentration of the substance was maintained at a relatively constant level, ensuring the necessary air exchange. To ensure this, we used an original inhalation (exposure) system that allows us to accommodate up to 10 small laboratory animals simultaneously. The system is a sealed exposure chamber equipped with a nebulizer with an aerosol sensor, inlet, and discharge pipes with special degassing traps. 

The mortality of animals in experimental groups was monitored daily. After the 8 days of treatments, the survived animals were euthanized by placing them in a CO_2_ chamber. The animals were then sent for dissection and pathomorphological examination.

### 4.10. Histology

On the 8th day of the study, the surviving animals from all groups were sacrificed, followed by lungs necropsy for macroscopic description, determination of lung weight (g) and histological assessment of lung tissue.

Lungs were fixed in 10% neutral formalin, dehydrated in alcohols of ascending concentrations, and embedded in paraffin. Paraffin sections 5 µm thick were obtained using an SM 2000R microtome (Leica, Germany), stained with hematoxylin and eosin (H&E), and examined using a DM1000 microscope (Leica, Germany). 

Scoring was performed by grading as follows:Infiltration of exudate in the alveoli:
1—Weak (10% of the lung area);2—Moderate (25% of the lung area);3—Intermediate (50% of the lung area);4—Severe (>50% of the lung area);
Stroma edema and thickening of the alveolar septa:
1—Normal lung;2—Moderate (<25% of lung area);3—Intermediate (25–50% of the lung area);4—Severe (>50% of the lung area).


### 4.11. Analysis of Fibrin/Fibrinogen Deposition

Quantitative analysis of fibrin/fibrinogen expression in lungs was carried out using immunohistochemical staining of tissue sections with polyclonal rabbit antibodies against fibrinogen (Abcam, ab64238). The percentage ratio of fibrinogen-positive areas to the total area of the lung tissue was determined by the following equation:$${\%\;}_{\mathrm{fibrinogen}}=\left({\;S}_{\mathrm{fibrinogen}}/{\;S}_{\mathrm{lungs}}\right)\;*\;100\%$$
where %_fibrinogen_ was the percentage of fibrinogen expression regions, S_fibrinogen_ is the area of the fibrinogen-positive regions in the image, S_lungs_ is the area of the lung tissue in the image.

For immunohistochemical staining, 5 μm paraffin sections of lung tissue were deparaffinized using xylene, 96 and 70% alcohol and boiled for 20 min in a universal antigen retrieval reagent (Abcam, ab208572) in a microwave oven. A hydrogen peroxide blocking solution (Abcam, ab64218) was then applied to the entire tissue section surface for 10 min. Protein blocking solution (Abcam, ab64226) was used for 5 min at room temperature to reduce nonspecific background staining.

Sections were stained with primary rabbit polyclonal antibodies against fibrinogen (Abcam, ab34269) in a solution of 1% BSA in PBS overnight at 4 °C. Rabbit-specific IHC polymer detection kit HRP/DAB (Abcam, ab209101), consisting of amplifying and detecting solutions, was used for signal amplification. Tissue sections were then stained with a DAB chromogen (Abcam, ab64238) for 2 min, and the cell nuclei were stained with hematoxylin (Abcam, ab220365) for 5 min at room temperature. After every step, the sections were washed with TBS solution (Abcam, ab64204) 3–4 times. The preparations were finally mounted under cover glasses using mounting medium for IHC (Abcam, ab64230).

Light images were registered at Axio Scope A1 microscope (Zeiss, Germany) using 20× dry objective and ZEN 3.0 software (Zeiss, Germany). Quantitative analysis of fibrinogen expression in the obtained images was performed using the ImageJ Fiji software (version 1.2; WS Rasband, National Institute of Health, Bethesda, MD) based on the previously described protocol [53]. Briefly, images were split into separate channels for hematoxylin and DAB using the “Color deconvolution” command. The intensity limit was determined for the DAB channel to cut off the background noise without affecting the positive signal. Then, the percentage of positive areas in the image converted into pixels (“Area fraction”) was measured. The tissue ROI was selected on the initial image using the “Color Threshold” function, with the further calculation of its area in pixels. 

### 4.12. Statistical Analysis

Results are expressed as the mean and standard deviation (SD) or standard error of the mean (SEM). The data were checked for normal distribution using the Anderson–Darling test. Nonparametric tests were used to compare differences among small samples (*n* ≤ 10), the Mann–Whitney U test was used to compare differences between two independent samples (control vs. experimental), and the Kruskal–Wallis test with Dunn’s post hoc analysis was used to compare differences among more than two samples. The reliability of immunohistochemical data was assessed using Brown–Forsythe and Welch one-way analysis of variance (ANOVA) and the Games–Howell test (for groups with *n* > 50). Differences between groups were considered statistically significant at *p* < 0.05. All statistical analyses were performed using GraphPad Prism software version 9.2.0 for Windows (La Jolla, CA, USA). In the figure legends, *n* refers to the number of samples or mice.

## Figures and Tables

**Figure 1 ijms-23-03417-f001:**
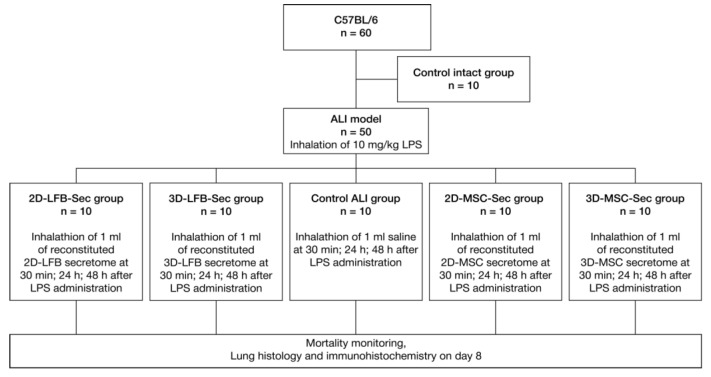
Flow chart of the study. ALI, acute lung injury; LPS, lipopolysaccharide; MSC, mesenchymal stromal cell; LFB, lung fibroblast.

**Figure 2 ijms-23-03417-f002:**
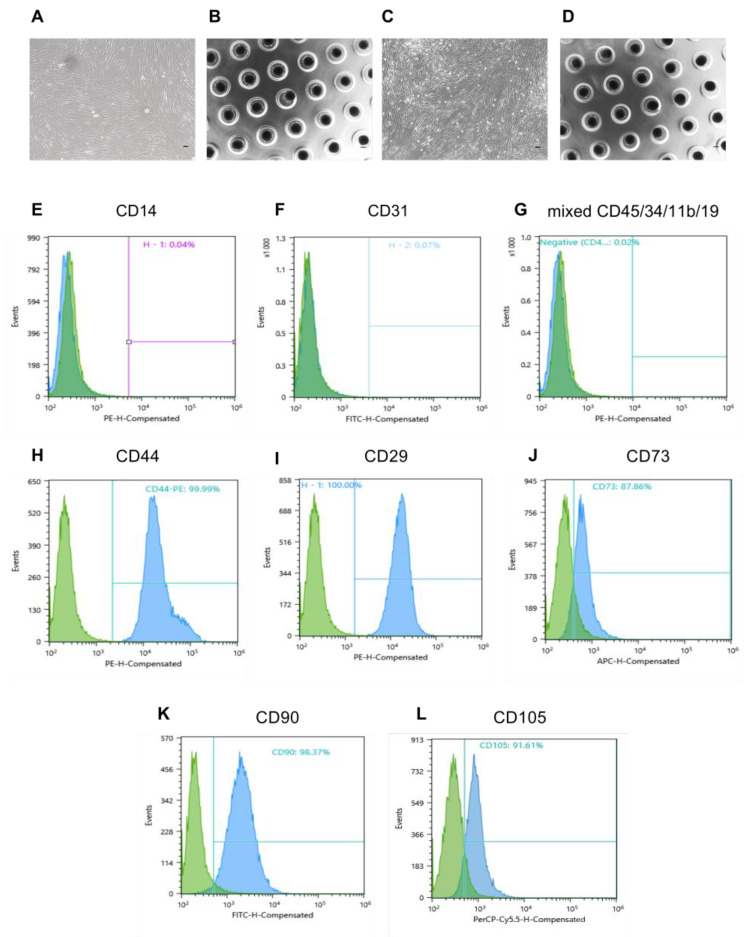
Morphology of MMSCs, lung fibroblasts and flow cytometric analysis of specific markers of MMSCs. (**A**) MMSCs at the third passage by day 3 in monolayer cultures. (**B**) MMSCs spheroids on day 3 in multiwell plates with agarose. (**C**) LFBs at the third passage by day 3 in monolayer cultures. (**D**) LFBs spheroids on days 3 in multiwell plates with agarose. Light phase contrast microscopy; scale bars = 100 μm (**A**,**C**) and 200 μm (**B**,**D**). (**E**–**G**) Negative endothelial and blood cell marker expression. (**H**–**L**) High expression of surface marker characteristic of MMSC population. MMSC, multipotent mesenchymal stromal cell; LFB, lung fibroblast.

**Figure 3 ijms-23-03417-f003:**
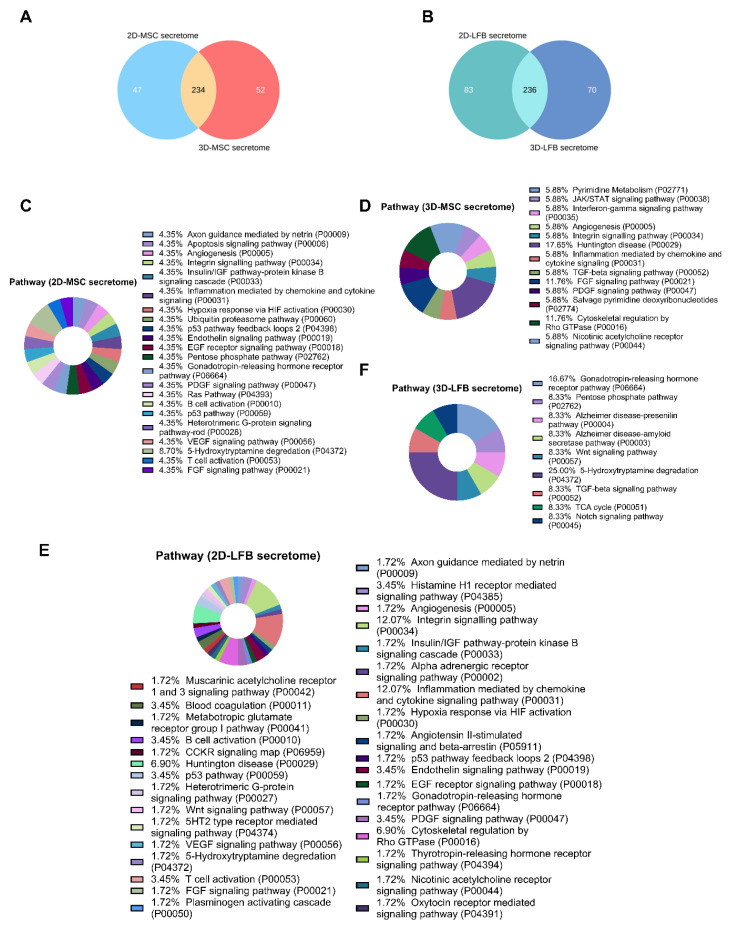
Proteomic analysis of multipotent mesenchymal stromal cell (MMSC) and lung fibroblast (LFB) secretomes. (**A**) Venn diagram of proteins identified in secretome from MMSC 2D/3D cultures. (**B**) Venn diagram of proteins identified in secretome from LFB 2D/3D cultures. (**C**) Gene Ontology pie chart of pathways associated with proteins identified only in the secretome from MMSC 2D cell culture. (**D**) Gene Ontology pie chart of pathways associated with proteins identified only in the secretome from MMSC 3D cell culture. (**E**) Gene Ontology pie chart of pathways associated with proteins identified only in the secretome from LFB 2D cell culture. (**F**) Gene Ontology pie chart of pathways associated with proteins identified only in the secretome from LFB 3D cell culture. MMSC, multipotent mesenchymal stromal cell; LFB, lung fibroblast.

**Figure 4 ijms-23-03417-f004:**
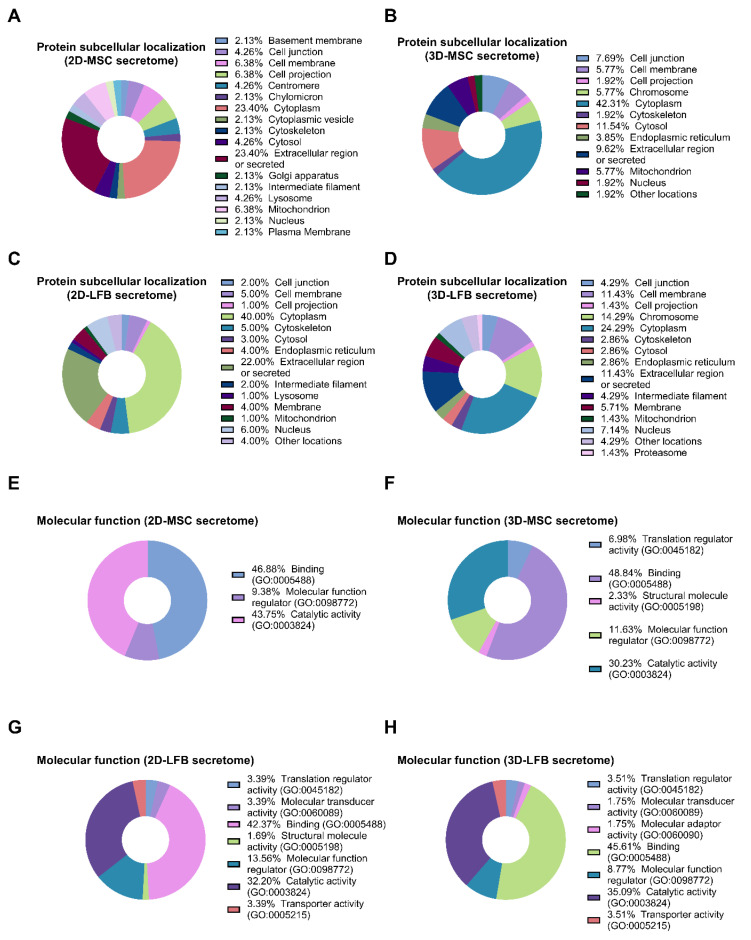
Proteomic analysis of multipotent mesenchymal stromal cell (MMSC) and lung fibroblast (LFB) secretomes. (**A**) Gene Ontology pie chart of subcellular localization of proteins identified only in the secretome from MMSC 2D cell culture. (**B**) Gene Ontology pie chart of subcellular localization of proteins identified only in the secretome from MMSC 3D cell culture. (**C**) Gene Ontology pie chart of subcellular localization of proteins identified only in the secretome from LFB 2D cell culture. (**D**) Gene Ontology pie chart of subcellular localization of proteins identified only in the secretome from LFB 3D cell culture. (**E**) Gene Ontology pie chart of molecular functions of proteins identified only in the secretome from MMSC 2D cell culture. (**F**) Gene Ontology pie chart of molecular functions of proteins identified only in the secretome from MMSC 3D cell culture. (**G**) Gene Ontology pie chart of molecular functions of proteins identified only in the secretome from LFB 2D cell culture. (**H**) Gene Ontology pie chart of molecular functions of proteins identified only in the secretome from LFB 3D cell culture. MMSC, multipotent mesenchymal stromal cell; LFB, lung fibroblast.

**Figure 5 ijms-23-03417-f005:**
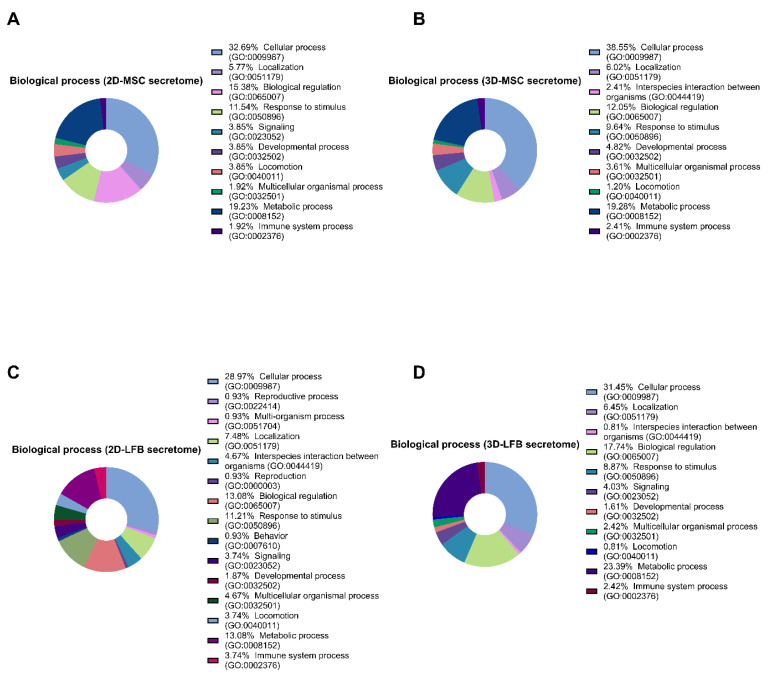
Proteomic analysis of multipotent mesenchymal stromal cell (MMSC) and lung fibroblast (LFB) secretomes. (**A**) Gene Ontology pie chart of biological processes of proteins identified only in the secretome from MMSC 2D cell culture. (**B**) Gene Ontology pie chart of biological process associated of proteins identified only in the secretome from MMSC 3D cell culture. (**C**) Gene Ontology pie chart of biological process associated with proteins identified only in the secretome from LFB 2D cell culture. (**D**) Gene Ontology pie chart of biological processes associated with proteins identified only in the secretome from LFB 3D cell culture. MMSC, multipotent mesenchymal stromal cell; LFB, lung fibroblast.

**Figure 6 ijms-23-03417-f006:**
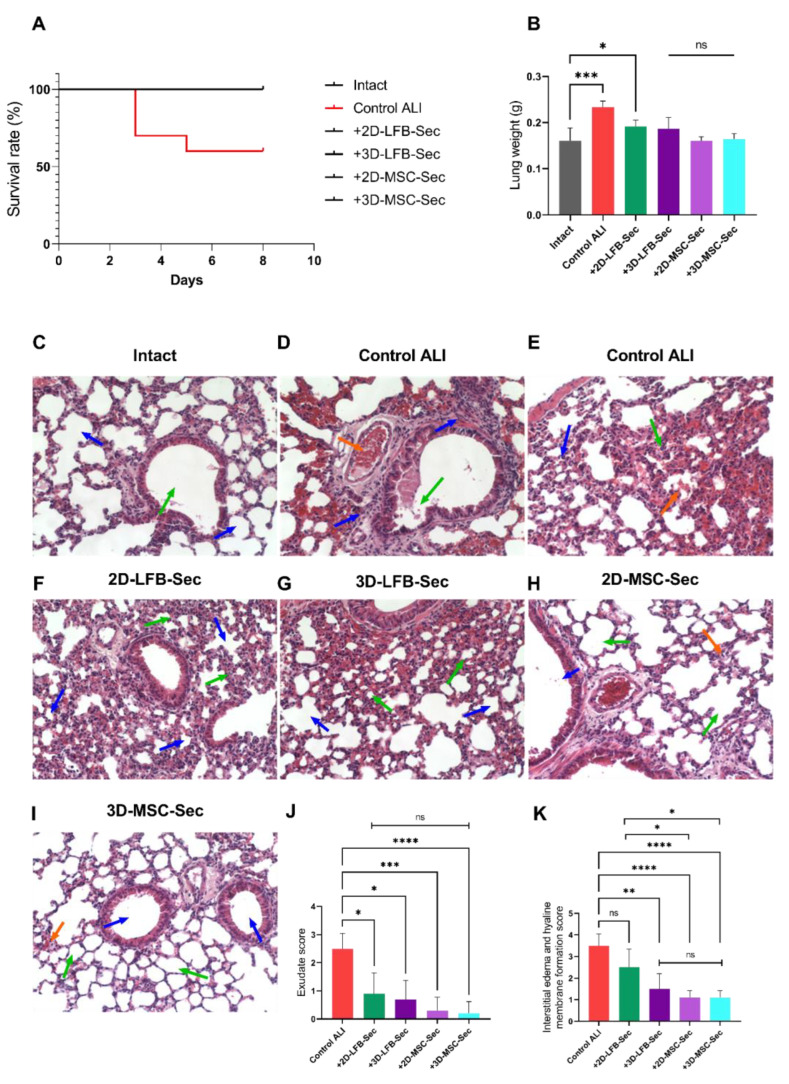
Survival analysis and histology examination of the lungs. (**A**) Survival of control, untreated lipopolysaccharide (LPS)-induced acute lung injury (ALI) mice, and LPS-induced ALI mice receiving 2D-LFB-Sec, 3D-LFB-Sec, 2D-MSC-Sec, and 3D-MSC-Sec for up to 8 days after LPS instillation (*n* = 10 for each group at the beginning of the experiment). (**B**) Lung weight on day 8 after ALI induction. Data are presented as the mean ± SD for each condition (*n* = 6–10). *****, adjusted *p* = 0.006 for Intact vs. Control ALI, *, adjusted *p* = 0.0323 for Intact vs. 2D-LFB-Sec, (ns) represents a nonsignificant difference for Intact vs. 3D-LFB-Sec, 2D-MSC-Sec and 3D-MSC-Sec. (**C**) Histology examination of the lungs for the Intact group. The blue arrows indicate the alveoli; the green arrow indicates bronchial lumen. (**D**,**E**) Histology examination of the lungs for the Control ALI group. The blue arrows indicate areas of infiltration and alveolar decline; the green arrows indicate epithelial disorders and edema of the stroma of the alveoli; orange arrows indicate accumulation of exudate and thrombosis. (**F**) Histology examination of the lungs for the 2D-LFB-Sec group. Blue arrows indicate the lumen of the alveoli; green arrows indicate the walls of the alveoli. (**G**) Histology examination of the lungs for 3D-LFB-Sec group. Blue arrows indicate the lumen of the alveoli; green arrows indicate the walls of the alveoli. (**H**) Histology examination of the lungs for the 2D-MSC-Sec group. The blue arrow indicates the lumen of the bronchi; green arrows indicate the lumen of the alveoli; the orange arrow indicates the wall of the alveoli. (**I**) Histology examination of the lungs for the 3D-MSC-Sec group. Blue arrows indicate the lumen of the bronchi; green arrows indicate the lumen of the alveoli; the orange arrow indicates the wall of the alveoli. Sections were stained with hematoxylin and eosin; ×200 magnification. (**J**) Exudate score in lungs. Data are presented as the mean ± SD for each condition (*n* = 6–10). (ns) represents a nonsignificant difference for multiple comparisons between 2D-LFB-Sec, 3D-LFB-Sec, 2D-MSC-Sec, and 3D-MSC-Sec. *** adjusted *p* = 0.0457 for Control ALI vs. 2D-LFB-Sec; *** adjusted *p* = 0.0115 for Control ALI vs. 3D-LFB-Sec; ***** adjusted *p* = 0.0002 for Control ALI vs. 2D-MSC-Sec; ****** adjusted *p* < 0.0001 for Control ALI vs. 3D-MSC-Sec. (**K**) Interstitial edema and hyaline membrane formation score in lungs. Data are presented as the mean ± SD for each condition (*n* = 6–10). (ns) represents a nonsignificant difference for Control ALI vs. 2D-LFB-Sec and multiple comparisons between 3D-LFB-Sec, 2D-MSC-Sec, 3D-MSC-Sec. ***, adjusted *p* = 0.0144 for 2D-LFB-Sec vs. 2D-MSC-Sec and 2D-LFB-Sec vs. 3D-MSC-Sec, ****, adjusted *p* = 0.0034 for Control ALI vs. 3D-LFB-Sec, ******, adjusted *p* < 0.0001 for Control ALI vs. 2D-MSC-Sec and 3D-MSC-Sec. All statistical analyses were performed using the Kruskal–Wallis test with Dunn’s post hoc test. LPS, lipopolysaccharide; ALI, acute lung injury; MSC, mesenchymal stromal cell; LFB, lung fibroblast.

**Figure 7 ijms-23-03417-f007:**
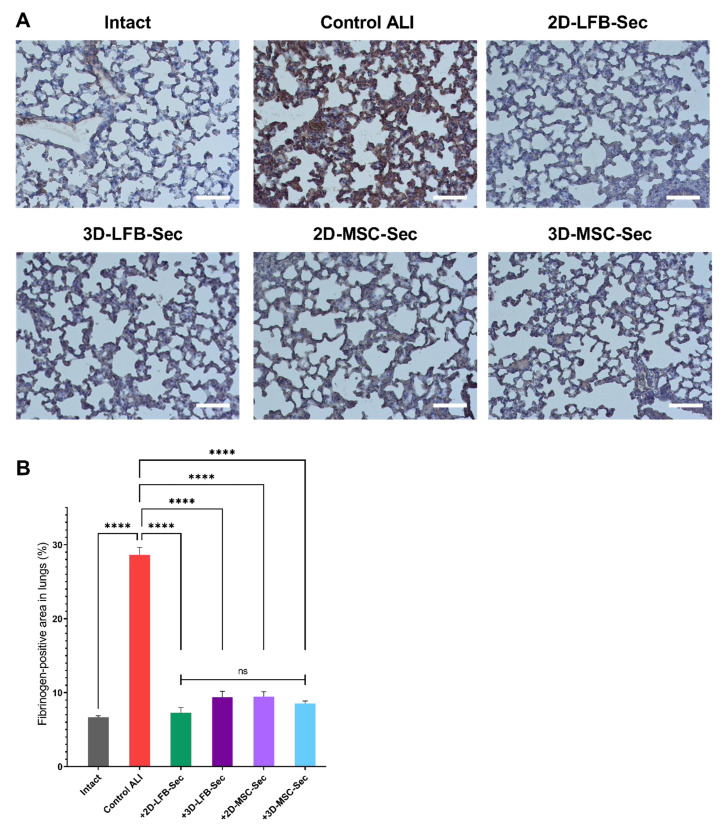
Immunohistochemistry and fibrinogen-positive area in lungs. (**A**) Lung tissue sections in the control, nontreated, and acute lung injury (ALI) groups after 2D-LFB-Sec, 3D-LFB-Sec, 2D-MSC-Sec, and 3D-MSC-Sec inhalation. Sections were stained with antibodies against fibrinogen, and cell nuclei were labeled with hematoxylin; ×20 magnification; scale bars = 100 μm. (**B**) The percentage of fibrinogen-positive areas in the total area of lung tissue in the control, nontreated, and ALI groups after 2D-LFB-Sec, 3D-LFB-Sec, 2D-MSC-Sec, and 3D-MSC-Sec inhalation. Data are presented as the mean ± SEM (*n* = 54). ****, adjusted *p* < 0.0001 for Intact vs. Control ALI, 2D-LFB-Sec vs. Control ALI, 3D-LFB-Sec vs. Control ALI, 2D-MSC-Sec vs. Control ALI, and 3D-MSC-Sec vs. Control ALI. (ns) represents a nonsignificant difference for multiple comparisons between 2D-LFB-Sec, 3D-LFB-Sec, 2D-MSC-Sec, and 3D-MSC-sec. Analyses were performed using Brown–Forsythe and Welch analysis of variance (ANOVA) with the Games–Howell post hoc test. ALI, acute lung injury; MSC, mesenchymal stromal cell; LFB, lung fibroblast.

## Data Availability

Additional data can be provided upon reasonable request from the date of publication of this article within 5 years. The request should be sent to the corresponding author at cd95@mail.ru.

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
