# Peer review of "Inhaled Placental Mesenchymal Stromal Cell Secretome from Two- and Three-Dimensional Cell Cultures Promotes Survival and Regeneration in Acute Lung Injury Model in Mice"

_ijms, 2022, doi:10.3390/ijms23073417_

Round 1
Reviewer 1 Report
The work of Kudinov et al. investigates the effect of inhaled secretoma of MMSC from placenta, produced in 2D and 3D, in an ALI mice model. Some important issues may be addressed:
Methods:
- Explain why the CM was produced during 72h. Previous studies justifying this data?
- What medium (and its composition) was used to produced CM?
- In what cell culture plate (multiwell (24, 12, 6?), petri, flasks (25, 75, 175…)) the CM has been produced?
- How many ml/cm2 for CM production? Based on previous data?
- Please, indicate references of antibodies from BD Biosciences (lines 358-360)
Results:
- Please, represent at least 2 representative areas for each condition in Figure 5-C.
- Indicate clearly in Figure 5 D and E, and state clearly in the text, if no significant differences was found between treated group.
- Idem for figure 6B
- Objective data (based on statistics) seem indicate no differences between LFB and CM, how do you explain that? Maybe some in vitro or molecular analysis will help in found different mechanism of action.
- In the abstract authors state: “In this study we investigated the proteomic profile of the secretome from 2D and 3D cultured placental MMSC and lung fibroblasts (LFBs) (…)” however the effect of the secretome of LFBs is not included in the abtract and not compared with secretome from MMSC. Also, they indicate: “We found that three inhaled administrations of freeze-dried secretome from 2D and 3D cultured placental MMSC protected mice from death, restored the histological structure of damaged lungs, and decreased fibrin deposition” but LFBs seem to show the same effect. Please, review the redaction of this part, because a relevant aspect of the results was not included (LFB’secretome effect).
Author Response
Dear Reviewer,
Thank you for your interest in our research and valuable comments.
We accept your advice and hope that the new version is better and we also would take it into account in our future studies.
The work of Kudinov et al. investigates the effect of inhaled secretoma of MMSC from placenta, produced in 2D and 3D, in an ALI mice model. Some important issues may be addressed:
Methods:
Point 1
Explain why the CM was produced during 72h. Previous studies justifying this data?
Response 1:
72h is indeed the most commonly used time for CM enrichment, based on previous studies (e.g., 10.1038/s41467-020-14344-7; 10.1016/j.lfs.2018.09.049)
Point 2
What medium (and its composition) was used to produced CM?
Response 2:
Full growth medium was used to produce CM. We acknowledge that this may be the limitation of our work, and cells should be cultured in minimal essential serum-free medium for CM collection. However, supplements such as FGF are significantly metabolized by cells and degraded within 72 hours, so we can assume that their therapeutic effect is minimized.
Point 3
In what cell culture plate (multi-well (24, 12, 6?), petri, flasks (25, 75, 175…)) the CM has been produced?
Response 3:
CMs from monolayer cultures were produced in 100-mm Petri dishes. Spheroids were cultured in multi-well agarose plates placed in the wells of 12-well plate (1 agarose plate with 256 spheroids per well of 12-well plate).
Point 4
How many ml/cm2 for CM production? Based on previous data?
Response 4:
The volumes of the produced CM are rarely stated in the literature. Presumably, they are standard volumes used in routine cell culture practice.
While it is easy to normalize the CM production rate on culture surface for monolayer cultures (10ml of CM from 100-mm dish, or 0.13ml/cm2), it is hard to do the same for 3D culture system with complex surface geometry. Thus, we normalized the volume of produced CM on the number of cells: we collected 10ml of CM from the final number of 2×106 cells (90-95% confluency). Precisely the same number of cells was transferred in 4 agarose plates (5×105 cells per plate, or ≈2000 cells per spheroid), and 2.5 ml of medium was added to each agarose plate. Therefore, we obtained the equivalent volumes of CM in 3D conditions.
Point 5
Please, indicate references of antibodies from BD Biosciences (lines 358-360)
Response 5:
Reference numbers have been added to the “Chemicals and reagents” section.
Results:
Point 6
Please, represent at least 2 representative areas for each condition in Figure 5-C.
Response 6:
The Information was added.
Point 7
Indicate clearly in Figure 5 D and E, and state clearly in the text, if no significant differences was found between treated group.Idem for figure 6B.
Response 7:
The indications were added. The manuscript has been corrected.
Point 8
Objective data (based on statistics) seem indicate no differences between LFB and CM, how do you explain that? Maybe some in vitro or molecular analysis will help in found different mechanism of action.
Response 8:
Thank you for your comment. We think that MSC secretome has shown a superior decreasing trend compared to other groups in terms of lung protection.
The goal of our pilot study was to explore the regenerative capacity of the secretome from 2D and 3D cultures in the ALI model. We agree that further in vitro and in vivo experiments are needed to elucidate the fundamental mechanisms that mediate the protective effects of secretome and its components in the settings of ALI.
We plan to conduct a study of molecular mediators and pathways responsible for the therapeutic effect of the secretome from various cell cultures in vitro on monocytes, alveolar epithelial cells, as well as to conduct a detailed study of the structure of the lungs in animals with an assessment of the effect on the recovery of alveolar epithelial cells I and II, the activity of enzymes with antioxidant action, as well as the evaluation of the functions of pulmonary surfactant in vivo.
Finally, in our ongoing research, we started profiling the differential miRNA expression in secretome vesicles in placental MSCs and lung fibroblasts, as miRNAs and mRNAs can have a significant impact on the therapeutic potential of the secretome. Of great interest is also the analysis of the lipid composition of vesicles secreted by various cell types from 2D and 3D cultures, as well as under preconditioning.
Point 9
In the abstract authors state: “In this study we investigated the proteomic profile of the secretome from 2D and 3D cultured placental MMSC and lung fibroblasts (LFBs) (…)” however the effect of the secretome of LFBs is not included in the abtract and not compared with secretome from MMSC. Also, they indicate: “We found that three inhaled administrations of freeze-dried secretome from 2D and 3D cultured placental MMSC protected mice from death, restored the histological structure of damaged lungs, and decreased fibrin deposition” but LFBs seem to show the same effect. Please, review the redaction of this part, because a relevant aspect of the results was not included (LFB’secretome effect).
Response 9:
Thank you for your comment. All secretome treated groups showed therapeutic effects in terms of survival, restoration of lung architecture, and decreasing fibrin reposition. MSC secretome exhibited superior therapeutic benefits than LFB-derived secretome in some measure. But we agree that LFB also demonstrated a high level of protection. The molecular mechanisms of these effects need to be studied in the future.
We didn’t point it in the previous version because the Abstract size is limited. We corrected the manuscript.
Reviewer 2 Report
In this article, authors proposed freeze-dried secretome derive from placental mesenchymal stromal cells for the treatment of acute lung injury mice model.
Overall, this study could be very interesting and could potentially open the way to the treatment of lung disease with MSC-derived secretome using a local administration. However, in my opinion, this study lacked some information. And also the design of the study does not considered some important aspects.
I think that some modifications and integrations should be added.
Comments:
- I suggest to add a graphical abstract or a figure with the presentation of design of the study, including the treatment groups of in vivo experiments. This could be useful foe the readers.
- The “full growth medium” used for MMSCs 2D and 3D culture contained ITS-G, FGF and heparin. Which is the rationale for the addiction of these compound to the medium?
- The “full growth medium” was used also for the secretome production? FCS contains many vesicles and growth factors. It is possible to distinguish vesicles produced by the cells and vesicles contained in the FCS?
- Each conditioned medium sample was only filtered with 0.22µm filter? No purification process was applied before the lyophilization? All components of culture medium were administered to the animals? The components of culture medium could induce any “therapeutic” effect? I think that a control group treated only with the culture medium (processed as for the conditioned medium) needs to be added to the study.
- Which is the aspect of the final lyophilized product?
- For the clinical use of lyophilized product it is necessary a scalable purification process. The presented approach is not scalable or usable on humans.
- No discussion was provided for the proteomic profile obtained by the 2D and 3D culture-derived secretome. The different content in terms of proteins could be very interesting and significant Why 3D-culture MMSC secretome contained different proteins? Based on proteomic profile, it is more advantageous the 3D culture?
- No information about the granulometric distribution of lyophilized product was reported. The particle dimension was one of the most important aspects for inhalation administration. I suggest to include in the future study a pharmaceutical technologist for the formulative study of the final products.
Author Response
Dear Reviewer,
Thank you for your interest in our research and valuable comments.
We accept your advice and hope that the new version is better and we also would take it into account in our future studies.

Reviewer 3 Report
Summary:
The authors evaluated the proteomic profile of the secretome from 2D and 3D cultured placental MMSC and investigated the therapeutic effects of lung fibroblasts (LFBs) and the effect of inhalation of freeze-dried secretome on survival, lung inflammation, lung tissue regeneration, fibrin deposition in LPS-induced ALI model. They found that three inhaled administrations of freeze-dried secretome from 2D and 3D cultured placental MMSC improved survival and attenuated LPS-induced lung injury.
General concerns:
- How to confirm that the inhaled secretome reach peripheral lungs and exert therapeutic effects?
- Results: The lung injury improvement was evaluated solely by histological examinations in this study. Please demonstrate lung injury improvement by measuring lung cytokines in lung tissues or bronchoalveolar lavage fluid or other markers.
- Materials and Methods: Data analysis, lines 517-518: Is it reasonable to compare Protein search was performed against “human amino acid sequences database” obtained from the UniProt KB (release May 2021) in a murine study?
- Figure 5C: Please confirm that two “Control ALI” figure panels are correct.
Author Response
Dear Reviewer,
Thank you for your interest in our research and valuable comments.
We accept your advice and hope that the new version is better and we also would take it into account in our future studies.
Dear Reviewer,
Thank you for your interest in our research and valuable comments.
We accept your advice and hope that the new version is better and we also would take it into account in our future studies.
The authors evaluated the proteomic profile of the secretome from 2D and 3D cultured placental MMSC and investigated the therapeutic effects of lung fibroblasts (LFBs) and the effect of inhalation of freeze-dried secretome on survival, lung inflammation, lung tissue regeneration, fibrin deposition in LPS-induced ALI model. They found that three inhaled administrations of freeze-dried secretome from 2D and 3D cultured placental MMSC improved survival and attenuated LPS-induced lung injury.
General concerns:
Point 1
How to confirm that the inhaled secretome reach peripheral lungs and exert therapeutic effects?
Response 1:
In our work, we used the classical inhalation chamber system to study the pharmacological activity of drugs administered by inhalation.
However, it should be noted that in such an administration model, on average, 50% of the drug enters the lungs (as a result of which such a large dose was inhaled), and the rest settle on the inner walls of the chamber and on the body of the animal, but since we used a validated ALI lethal model induced by LPS, at which the lethality is from 40 to 60%, one hundred percent survival in the treatment groups is a confirmation of the therapeutic effect. In addition, colleagues in a similar work have shown that the secretome administration using an inhalation chamber allows the formulation to reach the distal lung (Dinh, PU.C., Paudel, D., Brochu, H. et al. Inhalation of lung spheroid cell secretome and exosomes promotes lung repair in pulmonary fibrosis. Nat Commun 11, 1064 (2020). https://doi.org/10.1038/s41467-020-14344-7).
Point 2
Results: The lung injury improvement was evaluated solely by histological examinations in this study. Please demonstrate lung injury improvement by measuring lung cytokines in lung tissues or bronchoalveolar lavage fluid or other markers.
Response 2:
Thank you for your comment. We acknowledge this limitation.
Indeed, the key method for evaluating the efficacy is histological studies, and the absence of cytokine analysis in lavage or lung homogenate becomes a serious limiting factor.
Further studies are needed, especially on in vivo models of ALI, which will be the subject of our future research. In addition, we are going to do an immunohistochemistry assessment of some markers of lung recovery.
Point 3
Materials and Methods: Data analysis, lines 517-518: Is it reasonable to compare Protein search was performed against “human amino acid sequences database” obtained from the UniProt KB (release May 2021) in a murine study?
Response 3:
Thank you very much for your question. Rodents are often used to study cell preparations and acellular secretome/extracellular vesicles. A number of studies have already shown that many human proteins in the secretome are highly homologous with mice and rats.
Since in our work we used human cells to obtain a secretome, we turned to the human amino acid sequences database.
Point 4
Figure 5C: Please confirm that two “Control ALI” figure panels are correct.
Response 4:
We confirm that two “Control ALI” figure panels are correct. Thank you.
Round 2
Reviewer 1 Report
Point 1:
I expected previous data from the group that would justify the conditioning for 72 hours. If the authors refer to the literature, there are many more articles at 48h (some at 24h). Has no previous experiment been done, in vitro, by the group before the in vivo experiment?
48h: 10.14202/vetworld.2016.605-610 ; 10.1002/stem.1504 ; 10.1371/journal.pone.0225472 ; 10.1186/s13287-020-01782-9 ; 10.1016/j.biopha.2020.110584
24h: 10.1152/ajplung.00144.2011 ; 10.1186/s13287-016-0282-7
Point 2:
What happens regarding FCS? FBS, FCS and similar are known to contain extracellular vesicles (10.1002/jev2.12061; 10.1177/20417314211008626) which can influence on the MSC secretome observed effect. Have you used a control of CM without MSC?
The growth medium and the medium used for CM for LFB was not described. Is was the same than MMSC?
Point 6: Sorry, I did not ask for arrows (in that case, indicate in the figure legend also the arrows color definition). I ask to authors to show 2 representative areas of the tissue, I mean 2 pictures (different areas) corresponding to tissue section. Sorry for the misunderstanding.
Point 7: No change done regarding no significant data in new figure 7B.
Author Response
Dear Reviewer,
Thank you for your interest in our research and valuable comments.
We accept your advice and believe that it will make our future research better.

Reviewer 2 Report
Authors responded to all questions.
However, the problems presented in points 3 and 4 remain. Authors wrote that “the cells actively consume serum component” and “supplement such as FGF are significantly metabolized by the cells”.
I think that all components must be dosed after the culture to confirm these sentences.
In the most of scientific papers that presented the secretome production, serum-free medium was used for the secretome production and collection. I think that this is the better approach. However, I accept your idea and your approach.
Author Response

(The authors gave the same response as above.)

Round 3
Reviewer 1 Report
Dear authors, thank you for your answers but from my point of view they are not enough.
Point 1 was semi-clarified in the letter but not in the manuscript. My observations/requirements are made to aid readers' understanding, not just mine.
Point 2: Sorry, but this is a critical point because may be the serum is responsible of non significant differences and you do not use a control to solve this question.
Point 6: I do not understand why the new images were not added in a supplementary figure.
Author Response
Dear Reviewer,
Thank you for your interest in our research and valuable comments.
We appreciate your advice and your patience. We tried to correct all questions in the manuscript. Sorry for missing some important part in the previous version.
Point 1: was semi-clarified in the letter but not in the manuscript. My observations/requirements are made to aid readers' understanding, not just mine.
Response: The manuscript has been corrected. We added the explanation in the discussion part (lines 298-331).
Point 2: Sorry, but this is a critical point because may be the serum is responsible of non-significant differences and you do not use a control to solve this question.
Response: We acknowledge it. But cell culture conditions may influence on the secretome content and potential to. We tried to explain to the readers using of FCS without EV depletion procedure (lines 311-331). To minimize the effects of residual FCS-derived EVs and proteins we increased the time of conditioning and used identical full growth culture medium for all cells. Conditioned medium from LFB were used as controls. Using of growth media unconditioned by MMSC were also incorrect due to active consumption of growth factors and EVs by cells (lines 409-422). These data align with other studies showing that cells actively uptake soluble serum components and EVs for their growth and survival.
We believe that it has not yet been finally established which part of the secretome is more important - vesicles or soluble factors. Moreover, extracellular vesicles are quite heterogeneous, and separate populations of vesicles (exosomes, apoptosomes, etc.) can provide different therapeutic effects in vivo. This is the subject of further research.
Point 3: I do not understand why the new images were not added in a supplementary figure.
Response: Thank you for your advice. We have decided to add additional images to the Supplementary materials to avoid overwhelming the main manuscript and save the images size in Figure 6.
With best regards,
Vasily Kudinov